# Clinical utility of hepatitis C virus core antigen (HCVcAg) assay to identify active HCV infection in hemodialysis and renal transplant patients

Suresh Ponnuvel[1], Gnanadurai John Fletcher[1], Raghavendran Anantharam[1], Santosh Varughese[2], Vinoy George David[2], Priya Abraham[1]*

1 Department of Clinical Virology, Christian Medical College, Vellore, Tamil Nadu, India, 2 Department of Nephrology, Christian Medical College, Vellore, Tamil Nadu, India

* priyaabraham568@gmail.com

**Data Availability Statement:** All relevant data are within the manuscript and its Supporting Information files.

## Abstract

### Background

The prevalence of HCV infection is high and it is a major cause of liver-related morbidity and mortality in hemodialysis and renal transplant patients. Diagnosis of hepatitis C virus (HCV) infection requires both HCV antibody screening and confirmatory nucleic acid testing (NAT). Hepatitis C virus core antigen (HCVcAg) is a reliable direct viral marker to identify active HCV infection.

### Aim

To assess the clinical utility of HCV core antigen to identify active HCV infection in hemodialysis and renal transplant patients.

### Methods

A representative total of 231 plasma samples with a predominance of low viral load were included for HCVcAg testing and its performance characteristics were compared with the gold standard HCV RNA.

### Results

Comparison of HCVcAg with HCV RNA showed an excellent specificity of 99% (95% CI: 94.7 to 100%) and sensitivity of 80.62% (95% CI: 73.59 to 87.7%). Likewise, the PPV and NPV of HCVcAg were 99.1% (95% CI: 93.7% to 99.9%) and 80.2% (95% CI: 74% to 85.2%) respectively. The correlation between HCVcAg and HCV RNA was found to be good ($R^2$ = 0.86, p<0.0001). Among common Indian HCV genotypes (1, 3 & 4), good correlation was observed between HCV RNA and HCVcAg ($R^2$ = 0.81, p <0.0001).

### Conclusions

It is the first Indian study to show that HCVcAg is a reliable, cost-effective direct marker to identify active HCV infection in hemodialysis and renal transplant patients. Implementation

**Funding:** This study was supported by Fluid Research Grant of Christian Medical College, Vellore, Tamil Nadu, India (IRB grant No:12009). The funder had no role in study design, data collection and analysis, decision to publish, or preparation of the manuscript.

**Competing interests:** The authors declare that they have no competing interests.

of HCVcAg testing could improve the accessibility to efficacious and affordable disease management in hemodialysis and renal transplant patients. In HCVcAg negative cases, sequential testing with anti-HCV antibody followed by HCV RNA could be a reliable and cost-effective approach.

## Introduction

Globally, 80 million people are living with chronic hepatitis C virus (HCV) infection [1]. HCV infection is a significant cause of liver-related morbidity and mortality among hemodialysis and kidney transplant patients. The global prevalence of HCV infection in end-stage renal disease (ESRD) patients is 7.5% [2]. The serum HCV positivity in kidney transplant (KT) recipients is 10-fold higher than the general population [3]. In India, based on the anti-HCV and HCV RNA, the prevalence of HCV infection among hemodialysis patients is 4.3% to 45% and 27.7% respectively [4,5].

Patients with chronic kidney disease (CKD) are at high risk of acquiring HCV infection. Several high-risk factors have been identified for HCV infection among dialysis patients, which include prolonged vascular exposure, multiple blood transfusions, duration of end-stage kidney disease, mode of dialysis, and concurrent prevalence of HCV infection in the dialysis unit. Chronic HCV infection causes impaired health, increased risk of liver disease, higher cardiovascular mortality, and cryoglobulinemic syndrome in patients with hemodialysis [6,7]. In renal transplant recipients, HCV infection is independently associated with acute rejection, chronic allograft nephropathy, diabetes, and de novo glomerulonephritis [8,9].

The diagnosis of HCV infection is traditionally based on the detection of anti-HCV antibody and HCV RNA. However, the HCV antibody cannot be detected in the window period of 70 days [10]. Further, anti-HCV testing has not been reliable in dialysis patients because of the blunted humoral immune response that occurs with renal disease. The diagnostic limitations of antibody detection are that it cannot distinguish between current or past infection, and the false-negative rate is 17.9% in hemodialysis patients [11]. Thus, reliable direct markers such as HCV RNA or possibly HCV core antigen (HCVcAg) are needed to diagnose active HCV infection.

HCV RNA quantification is the recommended assay for the diagnosis, evaluation, and treatment by Kidney Disease Improving Global Outcomes (KDIGO) clinical practice guidelines [12]. HCV RNA is also used to assess treatment adherence and sustained virological response at 12 or 24 weeks after treatment completion. HCV RNA quantification by real-time reverse transcription-polymerase chain reaction (RT-PCR) is highly sensitive, specific, and reliable but expensive, time-consuming, and requires sophisticated equipment and trained personnel [13]. Though HCV RNA quantification is widely accepted as a gold standard, it is not suitable for routine screening among hemodialysis patients in resource-limited settings. Recently, HCVcAg quantification assay was developed, and its clinical sensitivity of 90% is comparable to HCV RNA. This assay was proved useful for the detection of early and active HCV infections in patients undergoing dialysis.

HCVcAg represents a robust stable marker of active infection than HCV RNA. It is released into the plasma during viral assembly, and it can be detected a few days after HCV RNA [14]. It is more stable than HCV RNA at room temperature, allowing easy transportation [15]. A positive HCVcAg result confirms the viral replication activity. In this study, we have ascertained the clinical utility of HCVcAg assay as a reliable, cost-effective, and high throughput

alternative to HCV RNA for the diagnosis of active HCV infection in hemodialysis and renal transplant patients.

## Methods

### Study design and patients

Ten mL of $K_2$EDTA blood samples were collected from hemodialysis and renal transplant recipients. Blood samples were centrifuged at 2500 rpm for 10 minutes for plasma separation and stored at -80˚C. Written informed consent was obtained from each patient before testing. We performed a study on 231 archived plasma samples from hemodialysis and renal transplant patients between January 2014 and December 2019. This study design was approved by the Institutional review board and ethics committee (IRB minute no: 12009) of the Christian Medical College, Vellore, India.

### Anti-HCV antibody estimation

Anti-HCV antibody was estimated by using chemiluminescent microparticle immunoassay (ARCHITECT HCV Ab, Abbott, Wiesbaden, Germany). The samples with <1.0 S/CO were considered negative. Samples with ≥1.0 to 5.0 S/CO and ≥5.0 S/CO were considered as weak positive and positive respectively. The weak positive samples were repeated in duplicates according to the testing manual.

### HCV RNA quantification and HCV genotyping

HCV RNA was quantified from plasma samples by using automated real-time PCR (Abbott Real-Time HCV RNA assay; Abbott Molecular, USA) with the dynamic range of quantification of 12 to $10^8$ IU/mL.

HCV RNA was extracted from plasma samples using QIAamp Viral RNA mini kit (Qiagen, Germany) as per the manufacturer's protocol. Sequencing of the HCV NS5b region was done using hemi-nested PCR. First-round PCR was performed using Titanium® One-Step RT-PCR kit followed by second-round PCR was performed using Super Therm Taq DNA polymerase (Medox®). The amplified 392 bp product from the second-round PCR was sequenced after a clean-up step to remove primer dimers and excess dNTPs. The nucleotide sequences obtained were subjected to BLAST using the HCV BLAST sequence database. The primer sequences are shown in **Table 1**.

### HCVcAg quantification

HCVcAg (ARCHITECT HCV Ag, Abbott, Wiesbaden, Germany) quantification was performed from the stored plasma samples. Plasma samples were centrifuged at 7000 rpm for 7 minutes and transferred (200 μL) into 2 mL sample cups to avoid any potential debris blocking

**Table 1. Primer sequences.**

| PCR | Primer name | Primer sequences (5'→ 3') | Target region |
|---|---|---|---|
| *First round PCR | Forward P1204 | GGAGGGGCGGAATACCTGGTCATAGCCTCCGTGAA | NS5b |
| | Reverse P1203 | GGGTTCTCGTATGATACCCGCTGCTTTGACTC | |
| *Second round PCR & *Sequencing PCR | Forward P1204 | GGAGGGGCGGAATACCTGGTCATAGCCTCCGTGAA | |
| | Reverse NS5b IP | TGATACCCGCTGCTTTGACTCNACNGTCAC | |

*Daniel et al. [16].

the Architect sample aspiration needle.The HCVcAg assay is a 2 step chemiluminescent microparticle immunoassay (CMIA) using microparticles coated with monoclonal anti-HCV for the quantification of HCVcAg. The first step of pretreatment lyses the viral particles and extracts the HCVcAg. In the second step, any HCV core antigen present in the pretreated sample binds to the anti-HCV coated microparticles conjugated with acridinium-labeled anti-HCV conjugate. The resulting chemiluminescent reaction is measured as relative light units (RLUs). The concentration of HCVcAg in each specimen was determined using an Architect HCV Ag calibration curve generated during the assay calibration. The linear range of HCVcAg quantification spans between 3.0 fmol/L and 20000 fmol/L. Samples with a concentration of >20000 fmol/L were diluted with automated dilution protocol. The assay cut-off threshold for a positive result was ≥3.0 fmol/L, whereas values of 3.0 to 10 fmol/L and >10 fmol/L were reported as weak positive and positive respectively. The weak positive samples (3.0 to 10 fmol/L) were repeated in duplicates as per the recommendation of the testing manual.

## Statistical methods

The sensitivity, specificity, positive predictive value (PPV), negative predictive value (NPV), and correlation (Spearman's rank correlation) of HCVcAg were calculated with the gold standard HCV RNA using MedCalc® v14.8.1 (Belgium) and GraphPad Prism software v8.0 (California). A p-value of <0.05 was considered statistically significant.

## Results

The HCVcAg quantification results of 231 plasma samples were compared with the gold standard HCV RNA for the calculation of sensitivity, specificity, correlation, PPV, and NPV. The demographic characteristics of enrolled patients are shown in Table 2 and the renal function profiles are shown in Fig 1.

   Among the 231, 129 (55.8%) samples were positive for HCV RNA and 102 (44.2%) samples were negative for HCV RNA. Among 129 HCV RNA positive samples, 33 (25.6%) samples had a lower viral load of < 3 log IU/mL. In the 129 HCV RNA positive samples, 99 (76.7%) samples were anti-HCV positive, and the remaining 30 (23.3%) samples were anti-HCV negative. In the 99 anti-HCV positive samples, 78 (78.8%) samples were positive for HCVcAg, and 21(21.2%) samples were negative for HCVcAg. Likewise, in the 30 anti-HCV negative samples, 26 (86.7%) samples were HCVcAg positive and 4 (13.3%) samples were HCVcAg negative. Among the 102 HCV RNA negative samples, 58 (56.9%) samples were anti-HCV positive and the remaining 44

**Table 2. Demographic characteristics of 140 enrolled patients.**

| Variables | n (%) |
|---|---|
| Median age (95% CI) | 36 (34–38) |
| Male | 106 (75.71%) |
| Female | 34 (24.2%) |
| HCV/HBV co-infected | 8 (5.71%) |
| HCV/HIV co-infected | 2 (1.42%) |
| HBV infected | 8 (5.71%) |
| HIV infected | 2 (1.42%) |
| Intravenous drug users | 2 (1.42%) |
| Liver transplant patients | 2 (1.42%) |

Abbreviations: n, number; CI, confidence interval; HCV, hepatitis C virus; HBV, hepatitis B virus; HIV, human immunodeficiency virus.

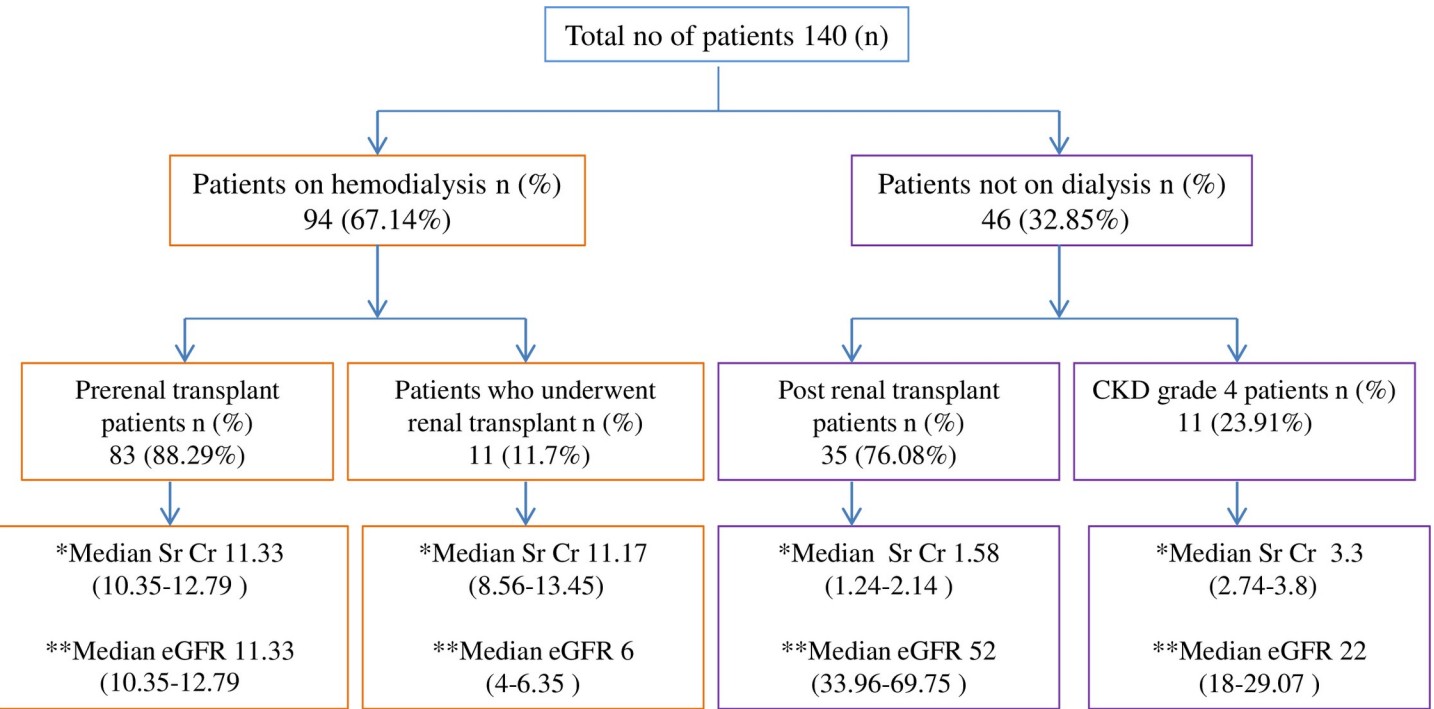

**Fig 1. Patients classification with renal function profiles.** *Median estimated glomerular filtration rate (eGFR) in ml/min/1.73m² with 95% confidence interval (p <0.0001). **Median serum creatinine (Sr Cr) in mg/dL with 95% confidence interval (p <0.0001).

(43.1%) samples were anti-HCV negative. In the 58 anti-HCV positive samples, 1 (1.7%) sample was positive for HCVcAg, and 57 (98.3%) samples were negative for HCVcAg. In the 44 anti-HCV negative samples, all the 44 (100%) samples were negative for HCVcAg (**Fig 2**).

The overall sensitivity, specificity, PPV and NPV of the HCVcAg were 80.62% (95% CI: 73.59 to 87.7%), 99% (95% CI: 94.7 to 100%), 99.1% (95% CI: 93.7% to 99.9%) and 80.2% (95% CI: 74% to 85.2%) respectively. The overall correlation between HCV RNA and HCVcAg was good ($R^2$ = 0.86, p<0.0001). The median HCV RNA was 1.9 log IU/mL (95% CI: 0 to 3.76 IU/mL), and the median HCVcAg concentration was 0 log fmol/L (95% CI: 0 to 0.73) (**Fig 3**).

In the HCV RNA positive samples (n = 129), a good correlation was found between HCV RNA and HCVcAg ($R^2$ = 0.8, p<0.0001). The median HCV RNA was 5.4 log IU/mL (95% CI: 4.99 to 5.71 IU/mL), and the median HCVcAg concentration was 2.5 log fmol/L (95% CI: 2.11 to 2.7) (**Fig 4**).

In samples (n = 99) that were positive for both HCV RNA and anti-HCV, a good correlation was found between HCV RNA and HCVcAg ($R^2$ = 0.77, p<0.0001). The median HCV RNA was 5.1 log IU/mL (95% CI: 4.79 to 5.69 IU/mL), and the median HCVcAg concentration was 2.2 log fmol/L (95% CI: 1.7 to 2.57) (**Fig 5**).

In samples (n = 30) that were positive for HCV RNA and negative for anti-HCV, an excellent correlation was found between HCV RNA and HCVcAg ($R^2$ = 0.91, p<0.0001). The median HCV RNA was 5.8 log IU/mL (95% CI: 5.31 to 6.15 IU/mL), and the median HCVcAg concentration was 3 log fmol/L (95% CI: 2.46 to 3.41) (**Fig 6**).

## Correlation between HCV RNA and HCVcAg in common genotype samples

Samples (n = 89) with three common genotypes (1, 3 & 4) showed good correlation between HCV RNA and HCVcAg ($R^2$ = 0.81, p<0.0001). The median HCV RNA was 5.4 log IU/mL

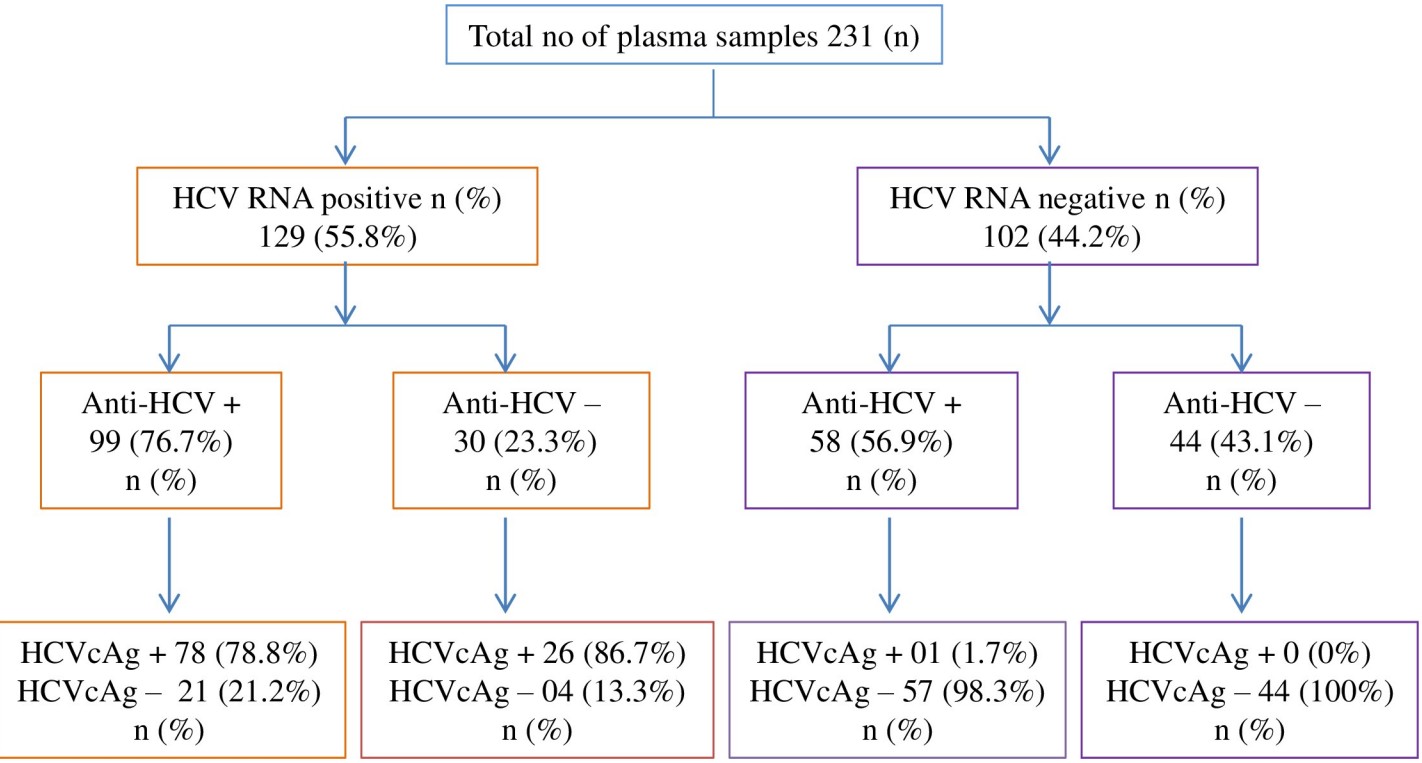

**Fig 2. Classification of samples based on assay results.**

(95% CI: 5.1 to 5.8 IU/mL), and the median HCVcAg concentration was 2.5 log fmol/L (95% CI: 2.27 to 2.7) (**Fig 7**).

### Receiver operating characteristic curve (ROC) analysis

The receiver operating characteristic curve provided the sensitivity of 100% and the specificity of 91.9% with the viral load of >3.39 log IU/mL, which is equivalent to 2500 IU/mL of HCV RNA (**Fig 8**).

## Discussion

HCVcAg and HCV RNA are the direct markers of HCV replication. However, there is a paucity of data on the utility of HCVcAg in hemodialysis and renal transplant patients. Such patients have a higher risk of HCV acquisition associated with disease management [5]. As the required frequency of HCV RNA testing is higher in this group, molecular testing escalates the overall cost by threefold when compared to the direct HCVcAg testing. Also, the reliability of anti-HCV antibody screening is not optimum among hemodialysis and renal transplant patients due to immunosuppression [17]. Hence, this study ascertained the utility of HCVcAg as an affordable viral marker to diagnose active HCV infection.

Samples for HCV RNA quantification are required to be frozen until the time of testing. Serum or plasma samples for HCVcAg do not require such specimen handling. This also makes it easier transportation to the nearest laboratory for HCVcAg estimation, and it can be tested on the same automated platform used for anti-HCV antibodies. The high positive predictive value of this HCVcAg permits screening of CKD patients before they start dialysis in a

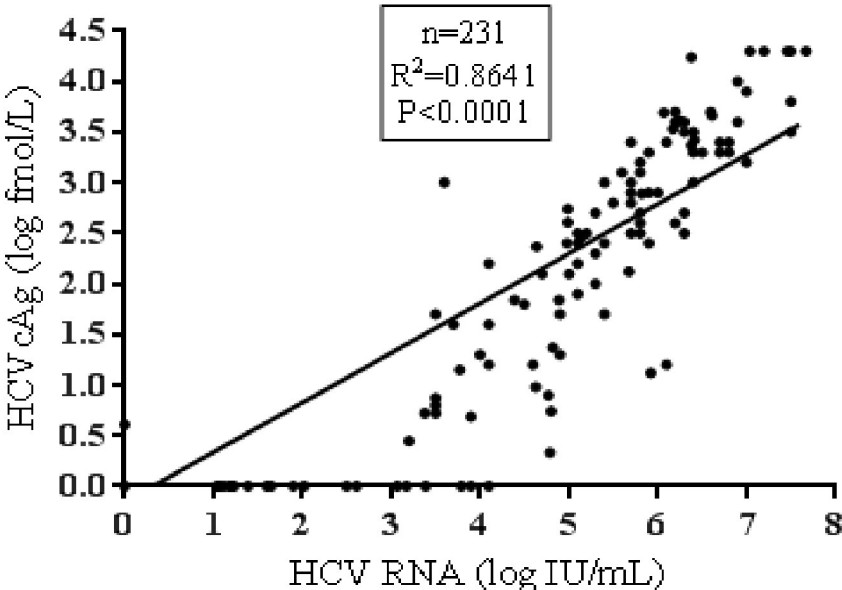

**Fig 3. Overall correlation between HCV RNA and HCVcAg.**

clinical setting. Due to the ease of testing and faster turnaround time, this assay can rapidly segregate infected and uninfected HCV patients.

This study demonstrated that the overall reliability and comparability of the HCVcAg with HCV RNA is good ($R^2$ = 0.86, p<0.0001). Comparing the HCVcAg assay with an entire range of HCV viral loads (1 to 7 log IU/mL) representing the natural course of HCV infection yielded $\geq$ 80% sensitivity. Analysis of HCV viral load data excluding $\leq$3 log IU/ml markedly

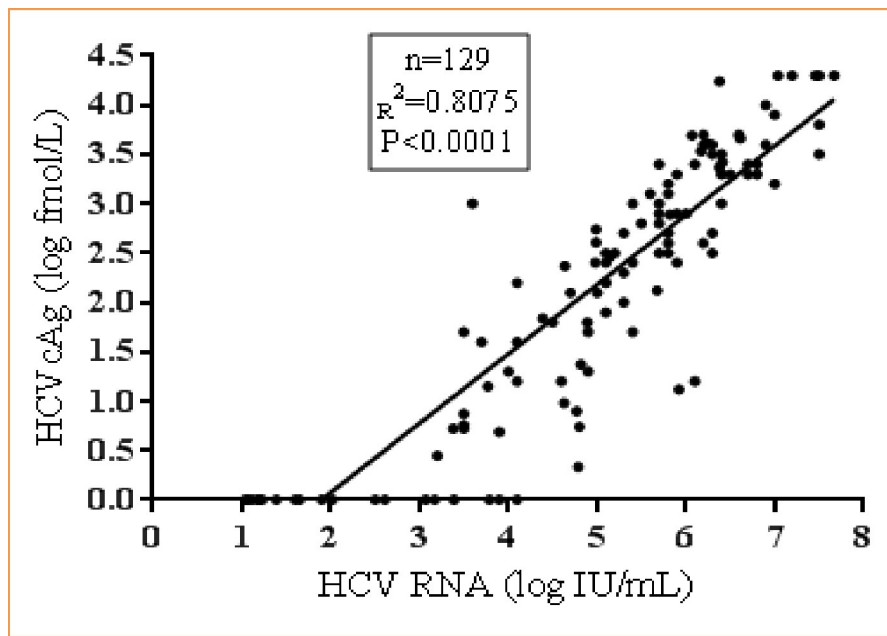

**Fig 4. Correlation between HCV RNA and HCVcAg in HCV RNA positive samples.**

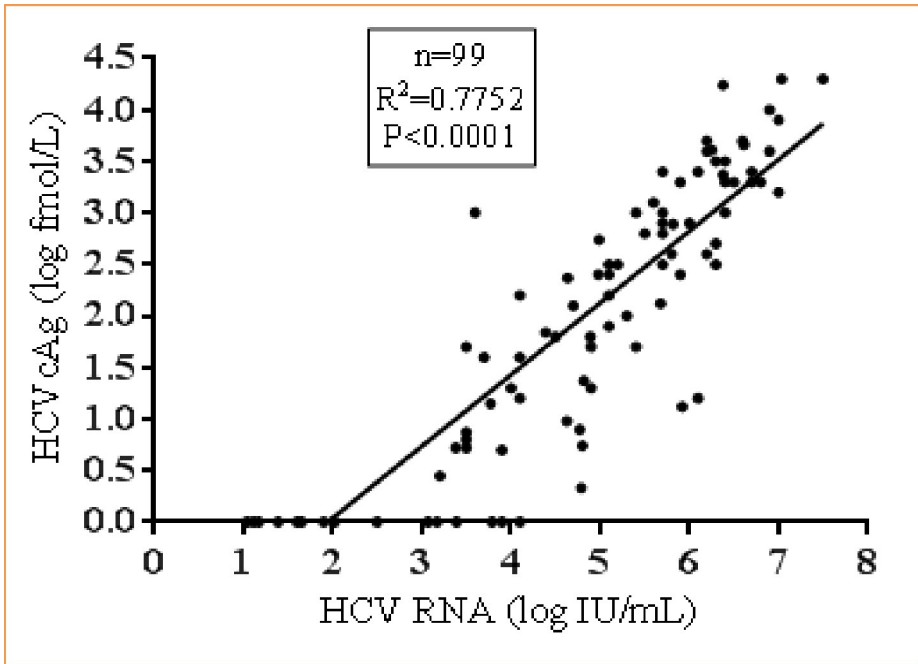

**Fig 5. Correlation between HCV RNA and HCVcAg in HCV RNA and anti-HCV positive samples.**

improved the overall sensitivity to 99% (data not shown), which is concordant with global data [18]. Typically in treatment naïve individuals, HCV viral load is in the range of $10^5$ to $10^6$ IU/mL [19]. A large cohort study (n = 10006) reaffirmed the clinical utility of HCVcAg as a screening/diagnostic assay to identify active HCV infection [20]. Furthermore, current study

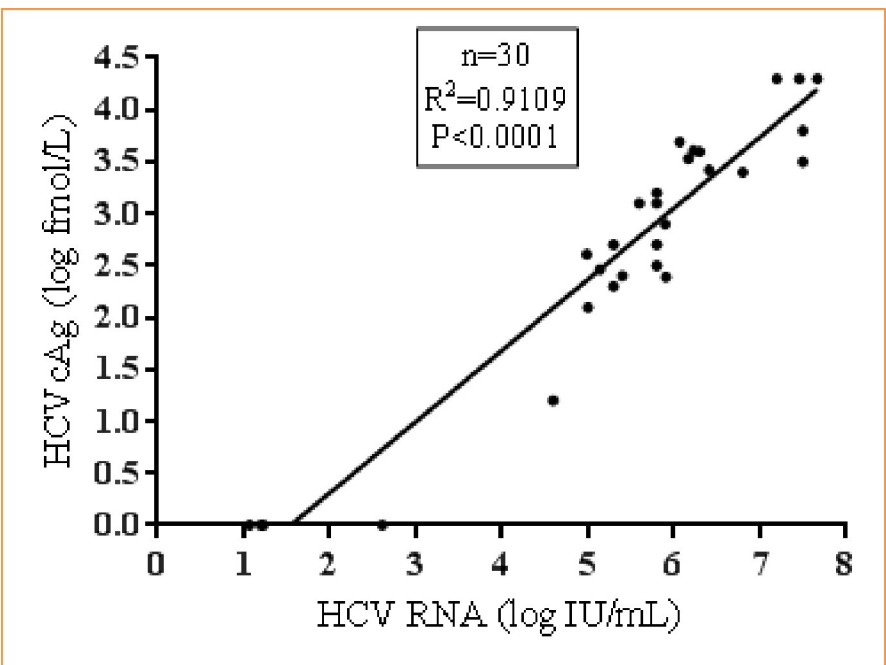

**Fig 6. Correlation between HCV RNA and HCVcAg in HCV RNA positive with anti-HCV negative samples.**

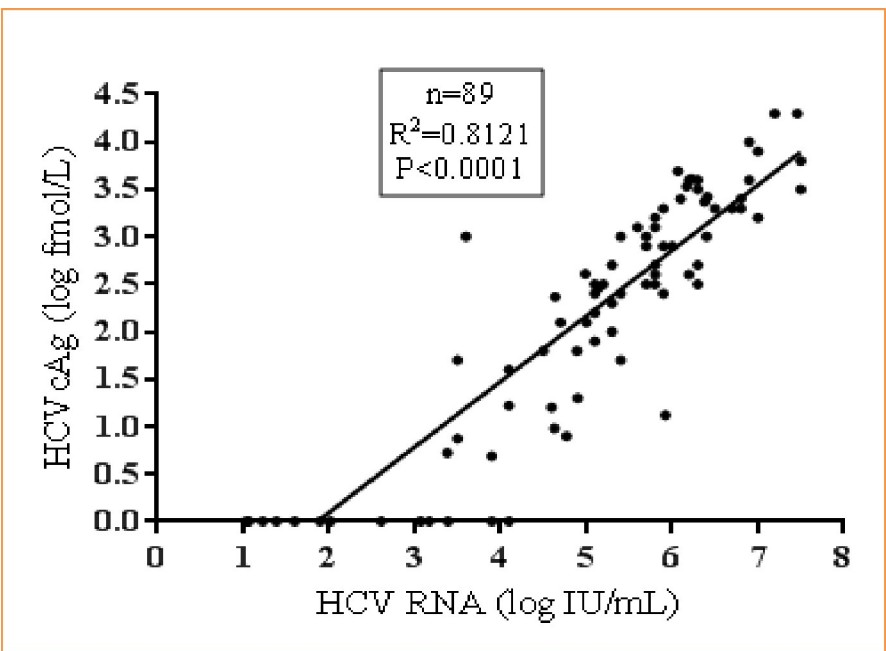

**Fig 7. Correlation between HCV RNA and HCVcAg in common genotype samples.**

data is concordant with EASL (2016) and the WHO Global Hepatitis Report (2017) recommendations to implement HCVcAg to identify active HCV infection [21,22].

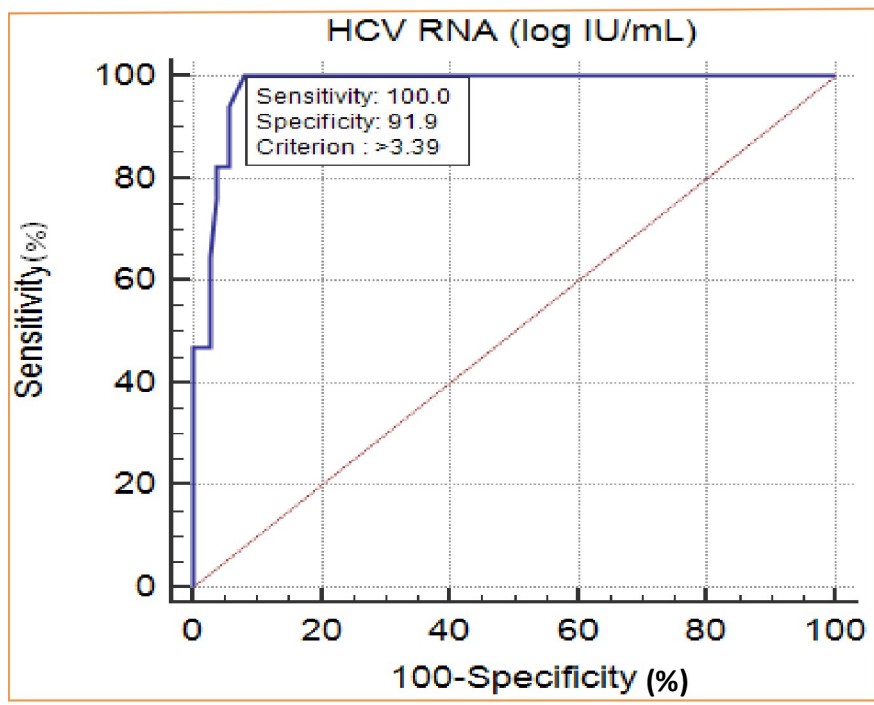

**Fig 8. Receiver operating characteristic curve (ROC) analysis.**

In anti-HCV negative with HCV RNA positive samples (n = 30), the HCVcAg assay showed an excellent correlation with HCV RNA ($R^2$ = 0.91). This can be attributed to the lack of measurable humoral response resulting in a higher HCV viral load than the anti-HCV positive patients. As the CKD group has a higher preponderance for immunosuppression leading to anti-HCV negative status, HCVcAg adds important diagnostic value in reliably identifying active HCV infection. Hence, this cost-effective assay with comparable performance characteristics with HCV RNA can be considered to identify active HCV infection.

A reliable direct HCV detection assay should be able to identify active infection caused by common genotypes. The current HCVcAg assay showed a good correlation with HCV RNA in hemodialysis and renal transplant patients infected with common genotypes (1, 3 & 4). However, the reliability of the assay needs further evaluation with other HCV genotypes.

One of the limitations of the study finding is the reduced sensitivity of HCVcAg assay with low-level HCV RNA samples, which showed a divergence of $10^3$ IU/mL. This discordance is expected because intrinsically PCR-based target amplification techniques are much more sensitive than signal amplification serological techniques [23]. To overcome this addition of polyethylene glycol (PEG) may be used during the HCVcAg assay which is a potentiator of Ag-Ab reactivity. This is achieved by steric exclusion of Ag and Ab by the PEG being more exposed in the dispersion medium and facilitating the Ag-Ab interaction [24]. HCV infection elicits strong anti-HCV core-antibodies, and there is a plausibility of HCVcAg and core-Ab complexes compromising the sensitivity of the HCVcAg assay cannot be ignored. Thus, dissociation of HCVcAg from anti-HCV antibodies may enhance the sensitivity of HCVcAg assay, which requires careful validation [25,26].

## Conclusions

HCVcAg is a sensitive and specific direct viral marker of active HCV replication. This assay is logistically easier, cost-effective, easy to perform, reproducible with a shorter turnaround time than HCV RNA testing. In low-resource settings, HCVcAg can be used as a confirmatory marker to replace HCV RNA testing in detecting active HCV infection among hemodialysis and renal transplant patients. In HCVcAg negative cases, sequential testing with anti-HCV antibody followed by HCV RNA could be a reliable and cost-effective approach. The clinical utility of this assay can be confidently extended to hemodialysis and renal transplant patients.

## Supporting information

**S1 File. Overall data.**
(XLSX)

## Acknowledgments

The authors thank the staff of viral hepatitis laboratory and department of nephrology for their valuable support in this study.

## Author Contributions

**Conceptualization:** Priya Abraham.

**Data curation:** Suresh Ponnuvel, Santosh Varughese, Vinoy George David.

**Formal analysis:** Suresh Ponnuvel, Gnanadurai John Fletcher.

**Funding acquisition:** Suresh Ponnuvel.

**Investigation:** Suresh Ponnuvel.

**Methodology:** Suresh Ponnuvel, Gnanadurai John Fletcher, Priya Abraham.

**Project administration:** Gnanadurai John Fletcher, Priya Abraham.

**Resources:** Santosh Varughese, Vinoy George David.

**Supervision:** Priya Abraham.

**Validation:** Gnanadurai John Fletcher, Priya Abraham.

**Visualization:** Suresh Ponnuvel, Gnanadurai John Fletcher, Priya Abraham.

**Writing – original draft:** Suresh Ponnuvel.

**Writing – review & editing:** Suresh Ponnuvel, Gnanadurai John Fletcher, Raghavendran Anantharam, Priya Abraham.

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
