## [Decision Letter · Decision Letter 0]

22 Dec 2020

PONE-D-20-35848

Clinical utility of hepatitis C virus core antigen (HCVcAg) assay to identify active HCV infection in chronic kidney disease patients

PLOS ONE

Dear Dr. Abraham,

Thank you for submitting your manuscript to PLOS ONE. After careful consideration, we feel that it has merit but does not fully meet PLOS ONE’s publication criteria as it currently stands. Therefore, we invite you to submit a revised version of the manuscript that addresses the points raised during the review process.

We look forward to receiving your revised manuscript.

Kind regards,

Chen-Hua Liu

Academic Editor

PLOS ONE

2.Please include your tables as part of your main manuscript and remove the individual files. Please note that supplementary tables (should remain/ be uploaded) as separate "supporting information" files.

Reviewers' comments:

Reviewer's Responses to Questions

**Comments to the Author**

1. Is the manuscript technically sound, and do the data support the conclusions?

Reviewer #1: Partly

Reviewer #2: Partly

2. Has the statistical analysis been performed appropriately and rigorously? 

Reviewer #1: Yes

Reviewer #2: Yes

3. Have the authors made all data underlying the findings in their manuscript fully available?

Reviewer #1: No

Reviewer #2: Yes

4. Is the manuscript presented in an intelligible fashion and written in standard English?

Reviewer #1: Yes

Reviewer #2: Yes

5. Review Comments to the Author

Reviewer #1: The authors investigated the clinical feasibility of the hepatitis C virus core antigen (HCVcAg) to detect active HCV infection in chronic kidney disease (CKD) pre/post-renal transplant patients. They recruited 231 plasma samples with predominant low HCV viral load and comparing HCVcAg testing with anti-HCV antibody and HCV RNA. They found that the specificity was 99% and sensitivity was 80.62% for HCVcAg comparing HCV RNA, and the PPV and NPV of HCVcAg were 99.1% and 80.2% respectively. They concluded that HCVcAg is a reliable and cost-effective marker in identifying active HCV infection in Indian CKD patients. Though not innovative, this is the first reported data applying HCVcAg on the detection of active HCV infection in pre/post renal transplant subjects in India. A few issues need further address or clarification for the version of this manuscript.

1. Clarify ‘CKD’ from ‘end-stage renal disease’ (ESRD) for this study population since pre/post-renal transplant including dialysis patients were enrolled and the authors did not give renal function data in the context and table 1.

2. How many patients received dialysis and what kind of dialysis (HD or CAPD) were they doing?

3. Did HD patients have more anti-HCV and HCV RNA discrepancy in this cohort?

4. How many patients had HCVcAg ‘weak positive’ especially in anti-HCV negative subjects in this cohort?

5. Give serum creatinine or eGFR data in table 1 and re-organize the structure of table 1, e.g. give n (%) instead of n/143, rename HBV/HIV infected to HBV/HIV co-infected and give HBV vs. HIV patient numbers respectively.

Reviewer #2: 1. Title: To put as CKD is "misleading". In actual fact, the study cohort was dialysis & transplant patients.

2. Introduction: Line 109, to say HCVcAg has comparable sensitivity as HCV RNA may not be fully accurate. Literature reported HCVcAg sensitivity is at best 90%.

3. Methodology: Line 146, it was not made clear by author if repeat testing was performed for sample which has values of 3 to 10fmol/l. This was recommended by the testing manual.

4. Results:

a. The author only emphasised on the correlation between HCVcAg & HCV RNA. This is not good enough is clinical setting. The clinicians would be interested to know the performance of HCVcAg in term of false positive or false negative. We cannot afford to miss any HCV + cases, esp. in dialysis settings.

b. Based on this analysis, the sensitivity of anti-HCV was 79.2% (99 out of 129) & HCVcAg was 80% (104 out 129). The sensitivity of both tests was almost the same. The author did not further elaborate on how to resolve the 20% false negative cases for HCVcAg.

c. The author also did not report the Anti-HCV & HCVcAg testing results in HCV RNA negative samples (N=102). This is important to make sure there is no high false positive.

5. Conclusion: It was not clear that if the author is advocating using HCVcAg as screening test to replace Anti-HCV OR using HCVcAg to replace HCV RNA in detecting active HCV infection. If it is former, the author needs to discuss if performing HCV RNA in all HCVcAg negative cases to rule out the 20% false negative cases will be cost-effective. If it is later, the author needs to justify if HCVcAg has the comparable sensitivity & specificity to HCV RNA to be used as confirmatory test, based on this study results. Sequential testing with Anti-HCV, followed by HCVcAg & HCV RNA only in HCVcAg negative cases will be a potential cost-effective approach to explore.

6. PLOS authors have the option to publish the peer review history of their article (what does this mean?). If published, this will include your full peer review and any attached files.

Reviewer #1: No

Reviewer #2: No

---

## [Author Response · Author response to Decision Letter 0]

4 Feb 2021

Cover letter, response to editor & reviewer, Manuscript with Track Changes, Manuscript, Figures and supporting information files has been separately uploaded as per the requirements mentioned in the decision letter.

Response to editor & reviewer 1: All the relevant data are available within the manuscript and its supporting information files

---

## [Decision Letter · Decision Letter 1]

23 Feb 2021

PONE-D-20-35848R1

Clinical utility of hepatitis C virus core antigen (HCVcAg) assay to identify active HCV infection in chronic kidney disease patients

PLOS ONE

Dear Dr. Abraham,

Thank you for submitting your manuscript to PLOS ONE. After careful consideration, we feel that it has merit but does not fully meet PLOS ONE’s publication criteria as it currently stands. Therefore, we invite you to submit a revised version of the manuscript that addresses the points raised during the review process.

We look forward to receiving your revised manuscript.

Kind regards,

Chen-Hua Liu

Academic Editor

PLOS ONE

Journal Requirements:

Reviewers' comments:

Reviewer's Responses to Questions

**Comments to the Author**

1. If the authors have adequately addressed your comments raised in a previous round of review and you feel that this manuscript is now acceptable for publication, you may indicate that here to bypass the “Comments to the Author” section, enter your conflict of interest statement in the “Confidential to Editor” section, and submit your "Accept" recommendation.

Reviewer #1: All comments have been addressed

Reviewer #2: (No Response)

2. Is the manuscript technically sound, and do the data support the conclusions?

Reviewer #1: Yes

Reviewer #2: Yes

3. Has the statistical analysis been performed appropriately and rigorously? 

Reviewer #1: Yes

Reviewer #2: Yes

4. Have the authors made all data underlying the findings in their manuscript fully available?

Reviewer #1: Yes

Reviewer #2: Yes

5. Is the manuscript presented in an intelligible fashion and written in standard English?

Reviewer #1: Yes

Reviewer #2: Yes

6. Review Comments to the Author

Reviewer #1: The authors response to my comments in the revised manuscript point-by-point accordingly though we have different opinion on definition of the study population (only CKD stage 4 & 5 were recruited in this study), I have no further comments.

Reviewer #2: The authors have responded to the previous comments but some are not reflected in the revised manuscript. I would like to highlight 2 comments which are not addressed satisfactorily in the revised manuscript as below:

1. "The author also did not report the Anti-HCV & HCVcAg testing results in HCV RNA

negative samples (N=102). This is important to make sure there is no high false positive."

- In my opinion, it is not good enough to put the data in supplementary document. I would like to see this to be reported in main manuscript. It should be made clear to readers that HCV core Ag does not have high rate of false positive.

2. "It was not clear that if the author is advocating using HCVcAg as screening test to

replace Anti-HCV OR using HCVcAg to replace HCV RNA in detecting active HCV infection.

If it is former, the author needs to discuss if performing HCV RNA in all HCVcAg negative

cases to rule out the 20% false negative cases will be cost-effective. If it is later, the author needs

to justify if HCVcAg has the comparable sensitivity & specificity to HCV RNA to be used as

confirmatory test, based on this study results. Sequential testing with Anti-HCV, followed by

HCVcAg & HCV RNA only in HCVcAg negative cases will be a potential cost-effective

approach to explore."

- The concern on how to handle the 20% false negative cases was not addressed in the revised manuscript even though the authors responded to the comments. In the conclusion, the readers can only get the message from the authors that "In low-resource settings, HCVcAg can be used as a confirmatory marker to replace HCV RNA testing in detecting active HCV infection." If authors agree that sequential testing with Anti-HCV, followed by HCVcAg & HCV RNA only in HCVcAg negative cases is a potential approach, this should be discussed in the manuscript.

7. PLOS authors have the option to publish the peer review history of their article (what does this mean?). If published, this will include your full peer review and any attached files.

Reviewer #1: No

Reviewer #2: No

---

## [Author Response · Author response to Decision Letter 1]

2 Mar 2021

As per the suggestion of Reviewer 1, the the name of the study population was modified as hemodialysis and renal transplant patients instead of CKD

---

## [Decision Letter · Decision Letter 2]

22 Mar 2021

PONE-D-20-35848R2

Clinical utility of hepatitis C virus core antigen (HCVcAg) assay to identify active HCV infection in hemodialysis and renal transplant patients

PLOS ONE

Dear Dr. Abraham,

Thank you for submitting your manuscript to PLOS ONE. After careful consideration, we feel that it has merit but does not fully meet PLOS ONE’s publication criteria as it currently stands. Therefore, we invite you to submit a revised version of the manuscript that addresses the points raised during the review process.

We look forward to receiving your revised manuscript.

Kind regards,

Chen-Hua Liu

Academic Editor

PLOS ONE

Journal Requirements:

Reviewers' comments:

Reviewer's Responses to Questions

**Comments to the Author**

1. If the authors have adequately addressed your comments raised in a previous round of review and you feel that this manuscript is now acceptable for publication, you may indicate that here to bypass the “Comments to the Author” section, enter your conflict of interest statement in the “Confidential to Editor” section, and submit your "Accept" recommendation.

Reviewer #2: All comments have been addressed

2. Is the manuscript technically sound, and do the data support the conclusions?

Reviewer #2: Yes

3. Has the statistical analysis been performed appropriately and rigorously? 

Reviewer #2: Yes

4. Have the authors made all data underlying the findings in their manuscript fully available?

Reviewer #2: Yes

5. Is the manuscript presented in an intelligible fashion and written in standard English?

Reviewer #2: Yes

6. Review Comments to the Author

Reviewer #2: Thanks for the revision.

A minor amendment needed. In conclusion, "In HCVcAg negative cases, sequential testing with anti-HCV antibody followed by HCVcAg & HCV RNA could be a reliable and cost-effective approach." Should omit the wording of HCVcAg.

7. PLOS authors have the option to publish the peer review history of their article (what does this mean?). If published, this will include your full peer review and any attached files.

Reviewer #2: No

---

## [Author Response · Author response to Decision Letter 2]

1 Apr 2021

Reviewer #2: Thanks for the revision.

A minor amendment needed. In conclusion, "In HCVcAg negative cases, sequential testing with anti-HCV antibody followed by HCVcAg & HCV RNA could be a reliable and cost-effective approach." Should omit the wording of HCVcAg.

Response: As per the reviewers request the conclusion has been modified as follows:

1) In HCVcAg negative cases, sequential testing with anti-HCV antibody followed by HCVcAg & HCV RNA could be a reliable and cost-effective approach." 

2)The word HCVcAg was omitted in the conclusion of the manuscript.

---

## [Editor Report · Decision Letter 3]

5 Apr 2021

Clinical utility of hepatitis C virus core antigen (HCVcAg) assay to identify active HCV infection in hemodialysis and renal transplant patients

PONE-D-20-35848R3

Dear Dr. Abraham,

We’re pleased to inform you that your manuscript has been judged scientifically suitable for publication and will be formally accepted for publication once it meets all outstanding technical requirements.

Kind regards,

Chen-Hua Liu

Academic Editor

PLOS ONE

---

## [Editor Report · Acceptance letter]

12 Apr 2021

PONE-D-20-35848R3 

Clinical utility of hepatitis C virus core antigen (HCVcAg) assay to identify active HCV infection in hemodialysis and renal transplant patients 

Dear Dr. Abraham:

I'm pleased to inform you that your manuscript has been deemed suitable for publication in PLOS ONE. Congratulations! Your manuscript is now with our production department. 

Kind regards, 

on behalf of

Dr. Chen-Hua Liu 

Academic Editor

PLOS ONE